# Performance of Storm Overflows Impacting on Shellfish Waters in England

**Andrew Younger [1],\*, Simon Kershaw [1] and Carlos J. A. Campos [2]**

1 Centre for Environment, Fisheries and Aquaculture Science, Weymouth Laboratory, The Nothe, Barrack Road, Weymouth, Dorset DT4 8UB, UK
2 Healthy Oceans Group, Cawthron Institute, 98 Halifax Street East, Nelson 7010, New Zealand
\* Correspondence: andrew.younger@cefas.co.uk; Tel.: +44-130-5206-600

**Abstract:** Storm overflow (SO) discharges to shellfish growing waters are a concern for shellfish growers and may pose a health risk to consumers. We investigated the performance of permitted SOs with a shellfish water spill monitoring requirement against the design criterion of 10 spills per year (averaged over 10 years) used in England. Performance against this criterion over the period 2019–2021 differed between the five water companies whose data was analysed. Across all companies, over half of SOs spilled more frequently than this criterion (percentage greater than 10 spills: 52%, 55% and 57% for 2019, 2020 and 2021, respectively). The number of SOs with the monitoring requirement also differed considerably between the water companies and consequently some companies deal with a significantly higher 'burden' than others. The number of SOs spilling more than 100 times in a year also differed between water companies, from 0% (Wessex Water) to 16% (United Utilities). Discharges from SOs can lead to short-term reductions in water quality that may be missed by routine monitoring programmes using faecal indicator bacteria such as *E. coli*. Such discharges can lead to a higher incidence of norovirus in shellfish and thus potential illness in consumers. We conclude that site-specific impact assessments, supported by spill event-based monitoring, are required given the increased demands on sewerage networks from urban growth and climate change and the need to improve shellfish production area classifications.

**Keywords:** shellfish water protected area; asset management planning; public health; water and sewerage companies; microbiological pollution; estuaries; combined sewer overflow



## 1. Introduction

Storm overflows (SOs) are structures in sewerage networks that spill excess wastewater and rainwater during heavy rainfall events and act as "safety valves" to protect properties from overloaded sewers causing flooding and wastewater backing up into streets and homes [1]. These overflow structures occur at storm tanks, pumping stations, and combined sewer overflows on the network or at wastewater treatment works inlets, and are a common feature of sewerage networks in England and other parts of the world. In 2017, there were 17,684 permitted SOs in England, of which 89% discharged to rivers, 10% to coastal waters, and 1% to groundwater [2]. In England, the Environment Agency (EA) issues permits to water and sewerage companies (WaSCs) to allow SO discharges during periods of wet weather when the capacity of the sewerage networks is reached. The permits specify the maximum size of solids that can be released, requiring a screen to be maintained on the outlet plus a minimum storage capacity before the outlet is used [3]. The use of overflows during periods of dry weather is considered by the EA a breach of permit [3].

The frequency, duration and volume of SO discharges are determined by many factors, namely the volumes of rainfall-generated runoff, sewer blockages or mechanical/electrical failures in sewerage networks, infiltration of groundwater or rainfall in sewers, and incorrect settings at SOs or wastewater treatment works inlets [4]. In recent years, WaSCs

have invested strongly in event duration monitoring (EDM) and all WaSCs in England are now required to release EDM data for publishing to the general public in the form of annual spill summary data. Event duration monitors register the timing and duration of SO spills and enable the production of detailed performance reports which are supplied to the EA on an annual basis. Currently, approximately 12,700 SOs have event duration monitors; the remainder are due to be installed by the end of 2023 [5]. This will enable a full picture of SO performance across the country. Where SOs discharge to shellfish water protected areas designated under the Water Environment (Water Framework Directive) (England and Wales) Regulations 2017 (hereafter referred to as shellfish waters), WaSCs also seek to notify relevant parties (shellfish harvesters, harbour authorities) about spills. Ideally, spill notifications occur on a near real-time basis to mitigate any detrimental effects of the discharges on the environment and human health. A number of trials have been undertaken to supply EDM information to shellfish farmers via internet and mobile phone systems to assist farmers in making informed decisions on when and where it is safe to harvest shellfish [6]. The Environment Act 2021 requires that WaSCs report spills in near real-time and discussions are now underway via the Defra chaired 'Storm Overflows Taskforce' to determine how this will be implemented.

Storm overflows discharge a mixture of untreated or partially treated wastewater and stormwater to receiving environments. Consequently, they release a wide range of contaminants, including pathogens, nutrients, toxicants, floatable matter, suspended solids, and oxygen-demanding contaminants [7,8] and have been associated with a wide range of adverse effects on the physical, chemical, and biological integrity of the environment. From a microbiological quality perspective, SO spills can contain a wide range of bacteria, viruses, and protozoa [9,10]. Whilst these discharges are generally diluted with rainfall runoff, they can nevertheless contain levels of microbiological contamination similar to those of crude sewage [10,11] and therefore represent a significant health concern due to the presence of human pathogens [12,13]. SOs pose a shellfish safety risk because many harvesting areas are impacted by SO spills and some shellfish (particularly oysters) are often eaten raw or lightly cooked. The transmission of disease to humans from consumption of contaminated shellfish following wastewater spills has been documented in the literature [14,15] and viral disease outbreaks, principally associated with the consumption of oysters, have been reported in the UK [16,17].

There is an intense ongoing debate about SOs in the UK and government has responded with several initiatives to improve the situation, principally through requirements of the Environment Act 2021. A key stipulation of the Act is to require the government to publish a plan to reduce wastewater discharges from SOs by September 2022 and report to Parliament on the progress towards implementing the plan [1]. This plan has now been published [18]. Key elements of this are that by 2035, water companies must improve all SOs discharging into or near every designated bathing water and improve 75% of overflows discharging to sites protected for ecological value. By 2050, this requirement will apply to all remaining identified SOs. Overflows that are causing the most harm will be prioritised for remedial action to give most benefit and water companies will be expected to prioritise nature-based solutions/green infrastructure over grey infrastructure in their planning.

The EA policy for permitting sewage discharges impacting shellfish waters published in 2003 contains design criteria to mitigate the environmental impacts of SO discharges. The policy includes a criterion of 10 significant and independent spills per annum on average over 10 years for SOs that are identified for improvement [19]. This design criterion is considered necessary to:

- work towards meeting the guideline standard in shellfish waters (300 *Escherichia coli* MPN/100 g), as required by the Shellfish Water Protected Areas Directions 2016 [20];
- prevent deterioration of those waters that already comply with this standard;
- improve to class B the shellfish production areas classified under the EU Food Hygiene Regulations as class C or those where harvesting is prohibited; and

- ensure that the production areas that are class A or B under the Regulations do not deteriorate.

Storm overflow permitting and performance measures have been thoroughly discussed in the peer reviewed literature [21–23]. Specifically, these have covered issues such as the need for introduction of telemetry for flow measurement (spill event duration monitoring) to assess SO performance along with discussion of the methods of permitting of SOs across Europe which are generally based on spill frequency. The public health impacts of SOs have recently been reported [24]. As the reporting and publishing of spill data from SOs is a relatively new requirement, there is currently little in the peer reviewed published literature on the status of their performance. The UK Government's appointed regulator for the WaSCs acknowledges that the current level of spills from SOs is unacceptable [25]. A recent study applied machine learning to wastewater treatment plant flow, rainfall and telemetry alarm data and detected additional storm tank overflows to those that had been detected by EDM [26]. A report by WWF in 2017 highlighted the poor ecological status of many UK rivers and the contribution of SOs to this situation [27]. There have been various reports more recently in the media on SO spill performance, noting the large number and duration of spills. We are not, however, aware of reports of SO discharge performance specifically in relation to the 10 spills design criterion used for shellfish water impact assessments. In this study, we analysed publicly available data to investigate SO spill performance for assets impacting on shellfish waters in England. The spill data that we reviewed were recorded by event duration monitors distributed in areas serviced by five WaSCs i.e., all those WaSCs with SOs impacting shellfish waters. Our specific objectives were to determine if spill performance is meeting the 10 spills standard; if performance differs between WaSCs and if the performance observed (as judged by spill numbers) is of concern regarding shellfish water quality.

## 2. Materials and Methods

### 2.1. Storm Overflow Spill Data

The five WaSCs considered in our study are: South West Water, Wessex Water, Southern Water, Anglian Water, United Utilities (Figure 1). We obtained spill data in the form of annual spill summaries for the period 2019–2021 published by the Defra Data Services Platform [28]. In selecting data for our assessment, we only considered SOs that have a specific EDM requirement for shellfish waters and only the above 5 WaSCs manage overflows with this requirement. This equated to around 1000 SOs out of approximately 15,000 in total across the five WaSCs in England. The total number of shellfish water SOs across the water companies for which EDM data were reported differed between years (1077 for 2019; 1033 for 2020 and 980 for 2021). The dataset used in our assessment contains spill records for a total of 3093 SOs across the five WaSCs and three years (Table 1).

**Table 1.** Number of SOs reported across the five WaSCs by year.

| Year | Wessex Water | Anglian Water | Southern Water | United Utilities | South West Water |
|------|------|------|------|------|------|
| 2019 | 41 | 50 | 246 | 329 | 411 |
| 2020 | 41 | 50 | 225 | 306 | 411 |
| 2021 | 37 | 40 | 261 | 269 | 373 |

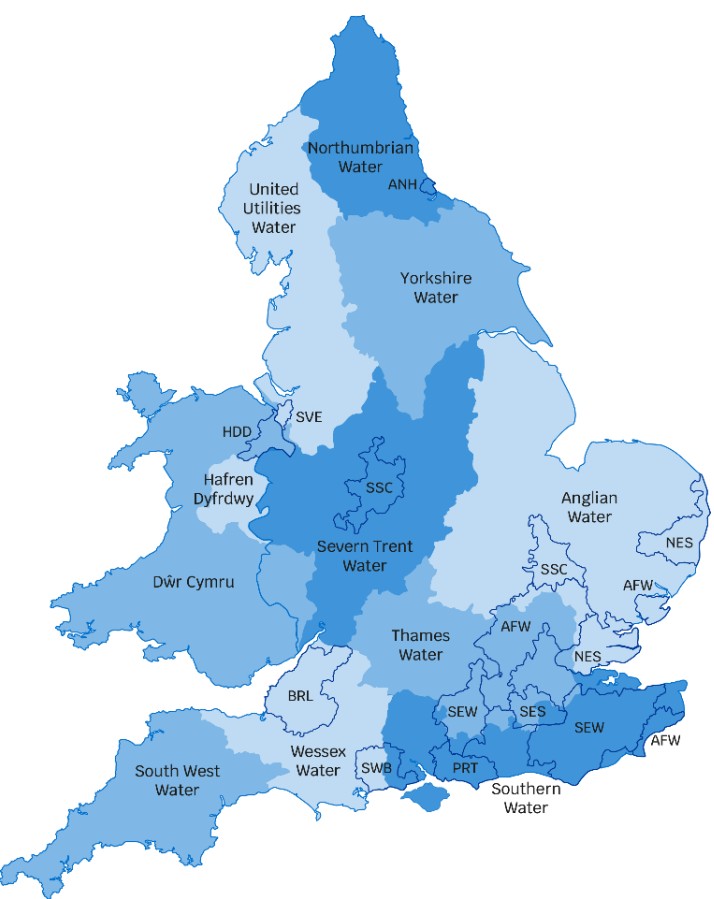

**Figure 1.** Areas serviced by water and sewerage companies in England and Wales. The present assessment is based on storm overflow data reported by South West Water, Wessex Water, Southern Water, Anglian Water and United Utilities. Source: Ofwat [29]. Key for abbreviated company names is available at: https://www.ofwat.gov.uk/households/your-water-company/contact-companies/ [accessed on 7 September 2022].

*2.2. Data Analyses*

We assessed the performance of SOs against the 10 spills design criterion. It should be noted that not all SOs have been designed to meet this standard as the EA allows another design option for SOs. This is a modeled water quality criterion of 1500 faecal coliforms/100 mL instead of the 10 spills criterion. The faecal coliform criterion applies to any area of the shellfish water over 97% of the time in the long-term. However, the intention of these two options is essentially the same in terms of water quality objectives so, for simplicity and to allow a direct comparison of performance, we used the 10 spill assessment for all SOs. According to the EA discharge permitting policy, the identification of SOs for improvement based on this criterion should be based on analysis of a 10-year averaged dataset [18]. However, the revised Urban Pollution Manual [30] and the guidance for Developing Spill Frequency Trigger Permits for SOs [31] do not explicitly mention this '10-year average'. A new method for counting spills (12/24 h block counting) was rolled out across England in 2018 and we understand that the EA places greater confidence in the spill data reported since then [32]. Therefore, we consider the use of a 3-year spill dataset (post-2018) appropriate and representative of SO performance. Because our assessment considers only SOs with a shellfish water event monitoring reporting requirement, not all WaSCs in England were included because not all have assets impacting on shellfish waters. The annual spill reports had total numbers of spills determined using the 12/24 h block counting method, which can be summarised as follows:

- Spill counting starts when the first discharge occurs;

- Any discharge(s) in the first 12 h block is counted as 1 spill;
- Any discharge(s) in the next and subsequent 24 h time periods are each counted as 1 additional spill per time period;
- The counting continues until there is a 24 h time period with no discharge;
- For the next discharge after the 24 h time period with no discharge, the 12 h and 24 h time period spill counting sequence begins again [32].

The spill profiles by year and water company were compared using Friedman tests and post-hoc pairwise Wilcoxon rank sum tests. All statistical analyses were conducted using the statistical software R [33].

## 3. Results and Discussion

### 3.1. Spill Profiles by Year and Water Company

Annual spill profiles (all WaSC data combined) are presented in Figure 2. The percentages of the total number of SOs across all WaSCs combined spilling ≤10 times over each year were as follows: 2019: 48.3%; 2020: 44.9% and 2021: 43.4%. These results suggest a slight deterioration in spill performance over the assessment period, however, this apparent deterioration was not found statistically significant either across all WaSCs (Q = 0.29, $p = 0.87$) or by WaSC ($p > 0.10$). It is important to note that the number of SOs reported with a shellfish waters EDM requirement differed considerably between WaSCs (ranging from 37 in the Wessex Water region to 373 in the South West Water region; based on 2021 figures) and therefore some companies have significantly more of a shellfish waters SO 'burden' than others. Nevertheless, the data show that the performance of SOs in England is, in many cases, falling short of that expected by the EA discharge consenting policy.

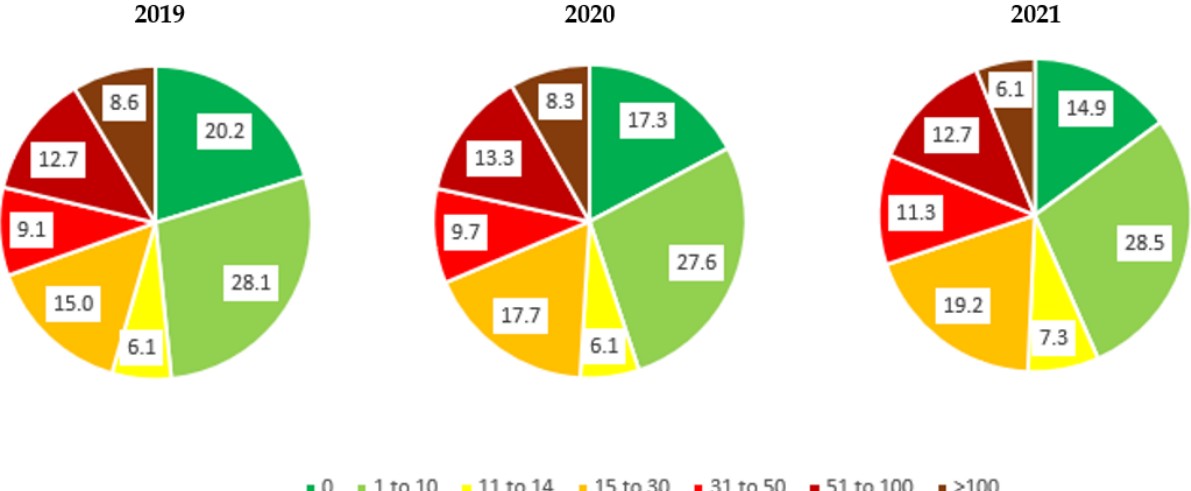

**Figure 2.** Storm overflow spill profiles for 2019, 2020 and 2021. Ranges are spill numbers. All water and sewerage company data combined.

Considering the 2021 data only, 43% of SOs met the 10 spills criterion. A further 7% met the threshold of 14 which is the value being considered by the EA as a new spill frequency trigger permitting criterion (it is proposed that this could be added into permits against which the measured spill performance could then be assessed) [31]. The remaining 50% exceeded the criterion, with 8% overall returning >100 spills each. The number of SOs spilling more than 100 times in a year differed between WaSCs from 0% (Wessex Water) to 15.8% (United Utilities). The maximum number of spills in any one year from one SO was 370 (United Utilities, 2019) while 15% of SOs for all five WaSCs combined recorded no spills at all in 2021. A study published by World Wide Fund for Nature-UK concluded from data provided by two WaSCs in the UK that 14% and 8% of overflows were spilling more than once a week and that 50% and 33% were spilling more than once a month [27]. Some overflows were spilling hundreds of times a year. The report also noted that, because SOs

are only allowed to spill in 'unusually heavy rainfall', it is likely that many were in breach of the requirements of the Urban Waste Water Treatment Directive. The spill frequencies reported by WWF are broadly in line with our own.

The SO spill performance was found to differ significantly between WasSCs each year at the 5% significance level (2019: Q = 14.16, $p$ = 0.028; 2020: Q = 20.31, $p$ = 0.002; 2021: Q = 19.38, $p$ = 0.004) and across all years (Q = 23.54, $p$ = $9.86 \times 10^{-5}$), with Wessex and Anglian Water having a statistically significant different SO spill performance than South West Water and United Utilities (i.e., generally performed better against the 10 spill standard) ($p$ < 0.039). Averaged over the 3 years for each of the five WaSCs, the seven spill categories used in our study are shown in Figure 3. The '1–10 spills' category is the most common category for all WaSCs. We identified a difference in spill performance across the WaSCs with United Utilities showing the widest range of spills and the highest spilling SOs overall. Wessex Water, with the lowest number of shellfish water SOs, returned the best performance overall with 79.8% of SOs spilling ≤10 times over the monitoring period. In general, the west of the country received more rainfall than the east over the 2019 to 2021 period [34]. The north west region in particular (served by United Utilities) consistently received above average rainfall during the monitoring period. This may at least partly explain the higher number of spills from SOs within this WaSC's network. However, the WaSC in the driest region (Anglian Water) was not the best performing company. Consequently, factors other than rainfall confound these results.

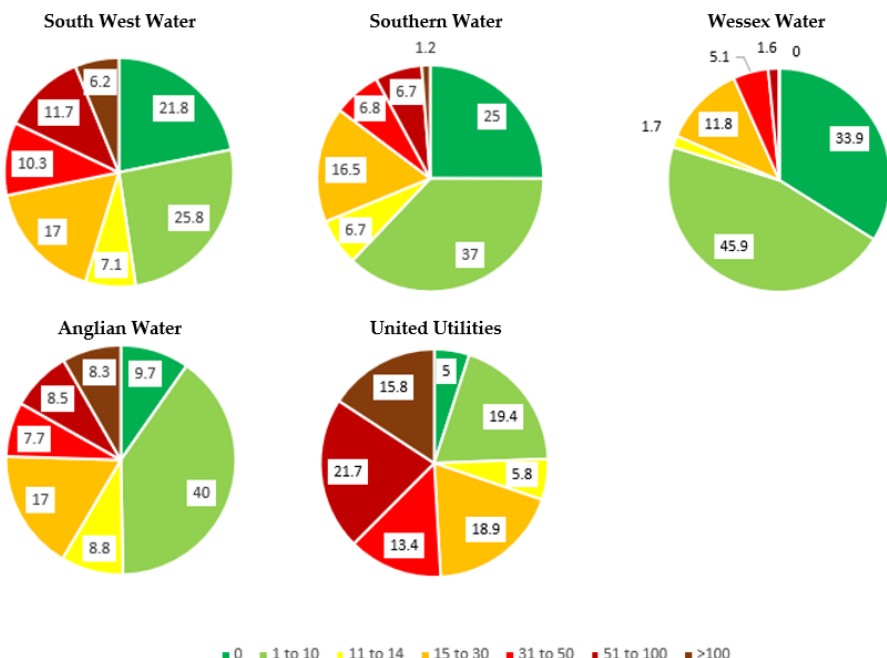

**Figure 3.** Storm overflow spill profiles for South West Water, Southern Water, Wessex Water, Anglian Water and United Utilities. Ranges are spill numbers, averaged for 2019–2021.

Our results broadly reflect the results of the limited number of published studies [27] in that a significant number of SOs appear to be spilling more than would be expected to achieve established environmental objectives. Recent media reports [35] highlight the large number and total duration of spills into shellfisheries with c. 29,000 spills into shellfisheries being recorded in 2022, lasting a total of 207,013 h—these data being obtained from published Government figures [28].

One factor that might have influenced these results is the ability of water companies to report spills. The study by Hammond et al. [26] mentioned above indicates potential for under-reporting of SOs, particularly in older sewerage networks more vulnerable to inflow and infiltration. Their study reports that between 2009 and 2020, the river stretch

downstream of the wastewater treatment work discharge may have received >360 spills lasting a whole day, often in extensive contiguous series of more than 10 days [26].

The 10 spills limit is a design criterion rather than a 'compliance' standard and, according to the EA discharge policy, is intended to be an average over 10 years. However, it is clear from our analysis that assets spilling more than 100 times have already exceeded the criterion if a 10-year dataset was used. Aside from any standards considerations, in terms of shellfish water quality and public health impacts, the number of spills in any particular year (i.e., rather than an average over 10) can be critical. This is further discussed in the next section.

### 3.2. Effects of Spills on the Microbiological Quality of Shellfish

It is a statutory requirement of Regulation (EU) 2019/627 [36] that commercial production areas be classified according to *Escherichia coli* standards in the shellfish flesh (European Commission, 2019). This classification determines what, if any, post-harvest treatment (e.g., purification, relay, or heat treatment) must be applied before the shellfish are sold for human consumption. Shellfish sampling targeted to spill events in a coastal embayment indicated that *E. coli* levels in shellfish flesh increase rapidly after the spill event to concentrations that may exceed 46,000 *E. coli* MPN/100 g (limit for 'prohibited' area classification status) and decay during subsequent dry, discharge-free periods [37]. During this bacterial accumulation period, *E. coli* concentrations can be concentrated in mussel tissues by more than 100 times the concentrations found in the surrounding water [37]. This indicates that the potential adverse impact of SO spills on harvesting area classification level is significant where these sporadic events happen to be detected in routine classification monitoring.

Whilst shellfish exposed to a single SO spill can take several days or even weeks to clear viral contamination to non-detectable levels [38,39], *E. coli* clearance from 'prohibited' area concentrations can take just a few days [37]. This highlights a key public health issue inherent to the classification system which is based on *E. coli* with, typically, monthly monitoring. Specifically, intermittent pollution such as that from SO spill events may be missed or inadequately represented. Where such situations arise, the classification category, and thereby the subsequent shellfish processing requirements, may be inadequate to protect consumers. However, the classification protocol for England and Wales contains several provisions for dealing with unusually high *E. coli* results, which may be associated with SO spills. When results are above the classification maxima, an investigation state is implemented. Results above certain trigger levels require an action state. In some situations, a temporary closure notice may be issued and complemented with additional safety measures (e.g., increased end-product testing, product withdrawal) [40].

A simple assessment of production area classification status against SO performance was considered in our study. However, we did not pursue this on the grounds that the outcome could be misleading. Essentially, it is not clear at this stage how significant the performance of SOs in individual areas is in terms of the harvesting area classification status (based on *E. coli* monitoring in shellfish) given all the other potential causes of pollution that may be relevant such as continuous wastewater discharges, run off from agricultural land, wildlife inputs and boating activity. A review of long-term bacteriological monitoring data for shellfish waters in Chichester Harbour showed log-order reductions in *E. coli* concentrations at sites closer to wastewater discharges following installation of a UV disinfection plant at a storm tank and other improvements to continuous treated discharges [41]. Despite the water quality improvements in the harbour channels, shellfish classification improvements were only observed in three out of eight beds. Because the *E. coli* test used for classification does not differentiate between animal and human sources of contamination and currently there is no requirement to monitor viruses in the classification programme (or the EU Regulations upon which this is based), individual area-specific studies with sampling targeting of spill events are necessary to clarify the SO pollution contribution in each case.

It is also a requirement of Regulation (EU) 2019/627 that commercial production areas be subject to an assessment of pollution sources of human and animal origins likely to impact on the waters in which the shellfish are cultivated ('sanitary survey'). Sanitary survey reports produced for English shellfish production areas show large numbers of SOs and other point sources of pollution in catchments draining to shellfish production areas.

Figure 4 shows only those permitted SOs in England with a shellfish waters associated spill reporting requirement, with summary spill data for 2021. The red hatched zones highlight the shellfish waters covered by each WaSC. (N.B. these do not represent the full extent of the area served by each WaSC.)

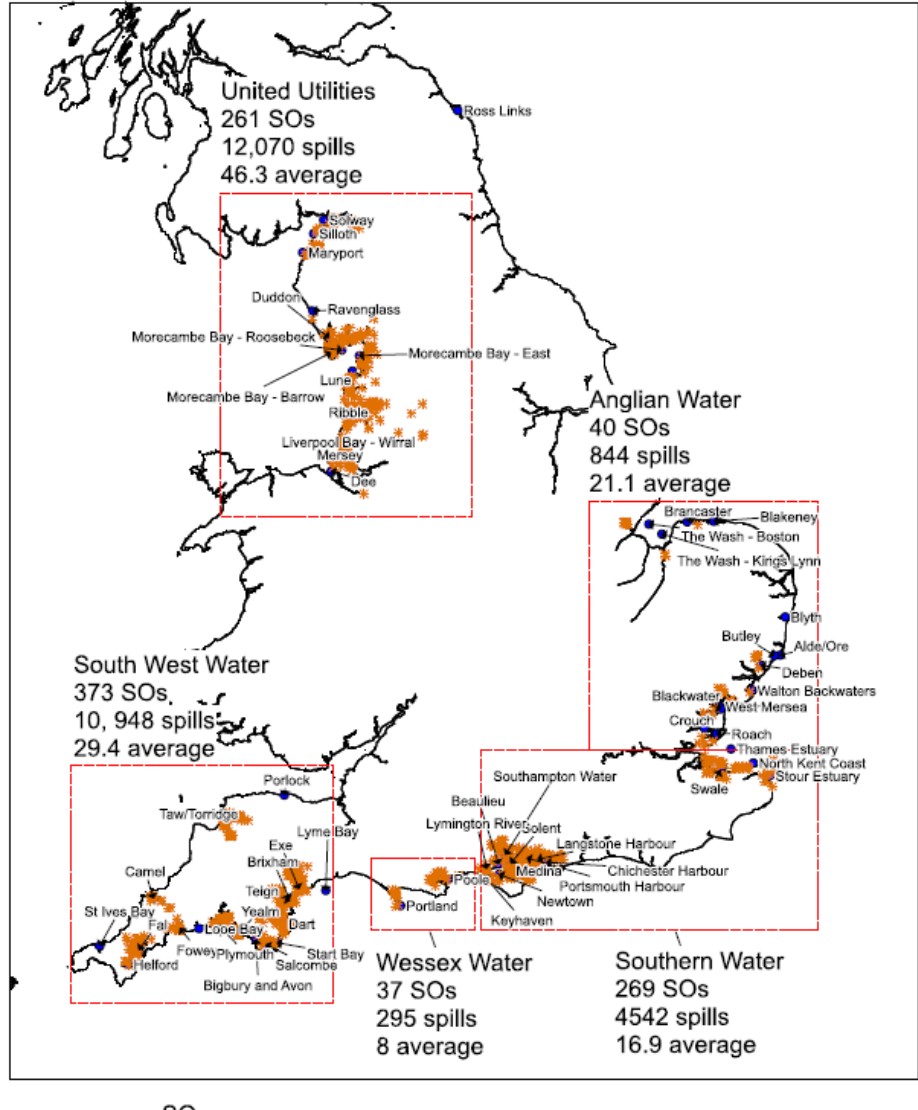

**Figure 4.** Permitted SO discharges in England with a shellfish waters-associated spill reporting requirement and classified bivalve shellfish production areas. Average figures represent the number of spills per SO by WaSC.

The need for growing shellfish in areas with good and stable water quality is clear. This need is especially important because the main post-harvest treatment used by the UK shellfish industry for class B shellfish (depuration) cannot be relied upon to remove enteric viruses from the shellfish [42]. The main virus of concern in relation to shellfish safety is human norovirus (NoV) [43]. In an English estuary, NoV was detected in oysters at concentrations of 1000 copies/g at 10 km downstream of a storm tank overflow [44,45]). These results are consistent with those in overseas studies indicating that NoV can contaminate growing waters that are several kilometers away from the discharges and persist in shellfish tissues for several weeks [39]. While the PCR methods used in these studies do not provide information on virus infectivity, in the event of a community outbreak infectious viruses will be discharged to the receiving environment because a single infected person sheds a high number of viruses and SOs discharge diluted untreated wastewater. Preliminary studies using both RT-qPCR to determine total concentrations and a plaque assay to determine infectious concentrations found that the proportion of infectious viruses (as predicted by FRNA bacteriophage GA) in SO discharges can be greater than that in UV disinfected effluents [46].

Previous research in our laboratory found a positive association between mean NoV levels in oysters sampled from classified production areas and the number of SOs in upstream catchments (Figure 5A). Our research also found a correspondence between the 10 spills design criterion and the limit of quantification of the NoV testing method (100 copies/g) [44,45] (Figure 5B). A comparison of NoV concentrations in oyster samples strongly linked to NoV outbreaks, with the levels typically found in commercial production areas, found virus concentrations in outbreak samples (GI + GII) ranging from 152 to 8215 copies/g [47]. Taken together, these results indicate that the NoV concentrations detected in commercial production areas are susceptible to causing illness and are also associated with the numbers of spills typically recorded in production areas.

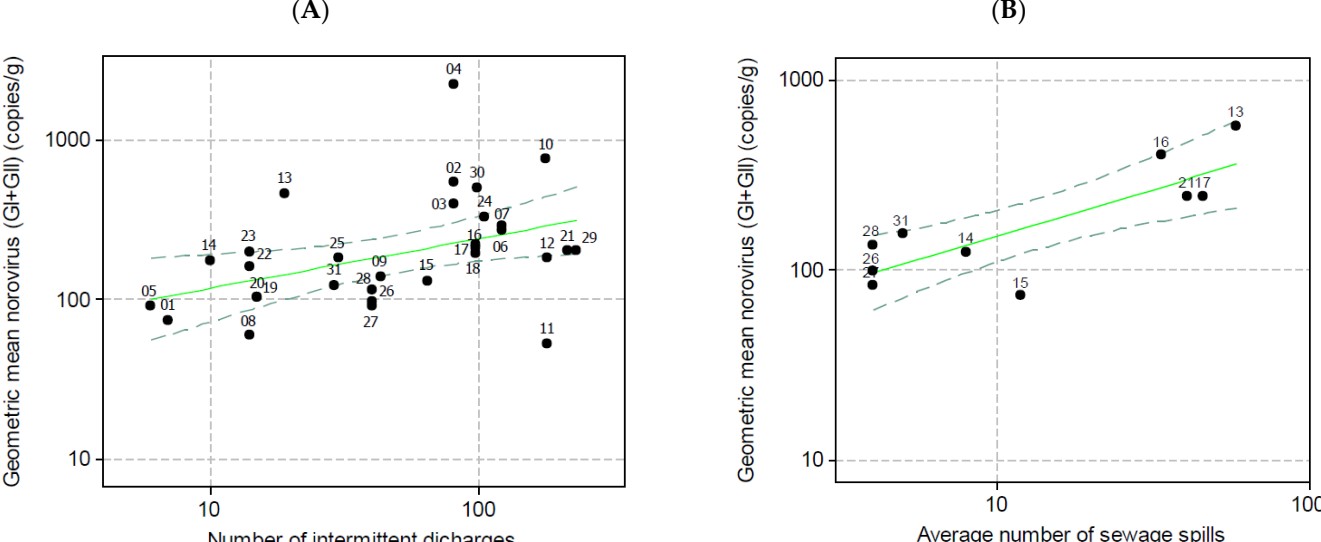

**Figure 5.** Linear relationships between mean concentrations of norovirus and the number of storm overflows (**A**) ($\log_{10}$geometric mean NoV = 1.758 + 0.3120 ∗ $\log_{10}$ number of discharges; $p < 0.019$; $R^2_{adj}$ = 15%)/average number of spills (**B**) ($\log_{10}$geometric mean NoV = 1.688 + 0.4904 ∗ $\log_{10}$ average number of spills; $p < 0.001$; $R^2_{adj}$ = 62%). Reproduced from Campos (2015) [45]. Numbers above black dots identify sampling sites as defined in Campos (2015). Continuous green lines represent linear regressions. Dashed green lines represent 95% confidence intervals.

The public health impacts from NoV are clearly a human wastewater issue but this may be associated with continuous discharges as well as SOs. The greater availability of EDM data, combined with upstream and downstream water quality monitoring studies

to assess the environmental impact of wastewater discharges (including SOs), as required under the Environment Act 2021 (monitoring yet to be implemented) will help clarify the contribution of SOs to production area classification and public health effects.

Alongside the potential health impacts are adverse financial effects on the shellfish industry. These can potentially be due to short or long-term closures or downgrading of beds, caused by SO-associated intermittent pollution of shellfish waters. Any closure or downgrading can incur potential loss of customers and/or increased production costs. In England, there is no up-to-date information on the economic losses from classification downgrades or reduced harvest access due to viral outbreaks or SO spills. However, overseas studies indicate these costs can be very significant. In Machias Bay (Washington County, ME, USA), Evans et al. [48] found that combined sewer overflows (CSOs) from the Machias wastewater treatment network was the largest single cause of soft-shell clam harvesting closures in the region and accounted for an estimated $2 million in forgone revenue during the period 2001–2009. The closures represented 17.5% of the total revenue from the clam fishery. We believe that the English shellfish industry faces similar challenges, with episodic poor water quality leading to product recalls and the consequent loss of consumer confidence impacting directly on the sustainability of farming operations and the reputation of the industry as a whole. Our argument is supported by the results of a recent consultation undertaken with a group of aquaculture stakeholders in Cornwall, Devon, Dorset and Somerset [49].

### 3.3. Future Perspectives

Following the substantial improvements to continuous discharges and the concomitant improvements in many English shellfish production area classifications since 2000 (AMP3) [50], SOs now represent a significant challenge in terms of their impact on microbial water quality and how this can be remedied in an affordable, cost-effective way. Reducing pollutant loads from SOs is challenging and expensive. Recent estimates indicate that the complete separation of wastewater and stormwater networks (eliminating SO spills) in England would cost between £350 billion and £600 billion [3]. This could increase household bills between £569 and £999 per year and is also highly disruptive and complex to deliver nationwide [3]. Complementary approaches such as sustainable urban drainage systems are increasingly required in new developments to help reduce wastewater spills entering waterways. A review of sewerage planning and drainage area plans undertaken by Atkins in 2011 [51] identified additional measures that could help achieve this objective:

- Sewer planning on a catchment/regional basis;
- More consistent use of risk-based approaches to assess the hydraulic capacity of the sewers;
- Development of confidence scoring systems for hydraulic models;
- More frequent use of real-time data as part of the operational management of sewer networks;
- Pro-active engagement in the planning process and more integrated asset planning systems.

Even if all these measures are implemented within the next few years, the increased demands on the sewerage network from urban growth and altered rainfall patterns associated with climate change will create more pressure on already stretched sewerage networks. Model simulations of spill volume, duration and frequency for 19 CSOs in North West England under climate change scenario predictions to 2080 showed an annual increase of 37% in total spill volume, 32% in total spill duration, and 12% in spill frequency for the shellfish water guideline standard [52]. Furthermore, quantitative microbial source apportionment modelling of NoV concentrations downstream of CSOs predicted increases by up to 24% under climate change scenarios which, if considered in relation to shellfish water quality, would represent an increased infection risk for this pathogen [53]. Under the Storm Overflows Discharge Reduction Plan [5], the UK government has set out ambitious targets for the next 30 years:

- By 2035, the environmental impacts of 75% of overflows affecting our most important protected sites will have been eliminated;

- By 2035, there will be 70% fewer discharges into bathing waters;
- By 2040, approximately 160,000 discharges, on average, will have been eliminated;
- By 2050, approximately 320,000 discharges, on average, will have been eliminated.

In the absence of statutory planning requirements for wastewater, the Water UK 21st Century Drainage Programme builds on the EA/Ofwat joint Drainage Strategy Framework and Ofwat recommendations to set the long term (25-year) direction, priorities, and pace for wastewater drainage activities. This will encourage greater consideration of future pressures such as growth, asset deterioration and climate change [54]. Whilst these are positive developments, we consider that there should be a need for an additional, longer-term strategy looking at 100 years and beyond.

## 4. Conclusions

Storm overflows play a vital role in preventing the flooding of homes and businesses when it rains. However, SOs can adversely impact shellfish water quality which is a concern for public health and the longer-term viability of shellfish farming businesses. Our assessment shows, for the first time in the peer reviewed literature, that a substantial number of SOs impacting shellfish waters are underperforming in terms of the shellfish water design criterion of 10 spills per year. The EDM data for 2021 reviewed in our study revealed that 43.4% of SOs with a shellfish water reporting requirement met this criterion. A further 7.3% met the threshold of 14 under the proposed Spill Frequency Trigger Permit criterion. The remaining 49.3% failed the criterion with 6.1% spilling >100 times each. The identification of poor performing overflows through the roll-out of EDM across the relevant SO networks is key to obtaining the necessary spill monitoring information to support site-specific investigations and pollution reduction plans for shellfish waters. As more EDM data become available, detailed assessments of site-specific SO impacts are needed to ensure that remediation measures can be targeted in the most cost-efficient way. We found that the contribution of SO spills to classification status and/or public health impacts in each shellfish production area is unclear at this stage but, based on the reported spill profiles, it can be substantial in some areas. Area-specific studies are required to clarify this contribution.

**Author Contributions:** Conceptualization, A.Y., S.K. and C.J.A.C.; methodology, A.Y., S.K. and C.J.A.C.; writing—original draft preparation, review, and editing, A.Y., S.K. and C.J.A.C. All authors have read and agreed to the published version of the manuscript.

**Funding:** This research was funded by the Cefas Seedcorn Programme under Project DP407A: 'Impact and evaluation of combined sewer overflow discharges in shellfish waters'.

**Institutional Review Board Statement:** Not applicable.

**Informed Consent Statement:** Not applicable.

**Data Availability Statement:** The spill data are available from the Defra Data Services Platform (https://environment.data.gov.uk/) [accessed 7 September 2022].

**Acknowledgments:** We are grateful to the Environment Agency for providing information and comments on an earlier version of this paper. We also thank Mickael Teixeira Alves (Cefas) for assistance with statistical analyses.

**Conflicts of Interest:** The authors declare no conflict of interest.

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
