# Peer review of "Performance of Storm Overflows Impacting on Shellfish Waters in England"

_land, doi:10.3390/land11091576_

Round 1

Reviewer 1 Report

The manuscript is well written and the subject is of great interest mostly considering the public health concern. A few small details should be corrected indicated in the specific  comments.

Specific comments:

Line 71: Never used abbreviations in the beginning of a sentence. In this case the authors should write SOs as Storm Overflows.

2. Material and Methods:

In this section the topics 3.1 (Line 106) and 3.2 (Line 14) should be 2.1 and 2.2 respectively. 

Lines 113-115: The number of SOs presented per year is not coherent with what is write bellow: in 2019 the SOs number is correct but for 2020 the correct SOs number is 1033 and not 1036 and for 2021 the correct t SOs number is 980, and not 1077. Please check this and correct accordingly.

3. Results and discussion:

Line 206: I think that this percentage is not correct. The correct number should be 79.8%. Please check this and change accordingly in the text.

Author Response

The manuscript is well written and the subject is of great interest mostly considering the public health concern. A few small details should be corrected indicated in the specific  comments.

Specific comments:

Line 71: Never used abbreviations in the beginning of a sentence. In this case the authors should write SOs as Storm Overflows.  Response: We have adjusted as suggested

  1. Material and Methods:

In this section the topics 3.1 (Line 106) and 3.2 (Line 14) should be 2.1 and 2.2 respectively.  Response: We have renumbered accordingly.

Lines 113-115: The number of SOs presented per year is not coherent with what is write bellow: in 2019 the SOs number is correct but for 2020 the correct SOs number is 1033 and not 1036 and for 2021 the correct t SOs number is 980, and not 1077. Please check this and correct accordingly.  Response: thank you for spotting this(!)  We have adjusted as appropriate.

  1. Results and discussion:

Line 206: I think that this percentage is not correct. The correct number should be 79.8%. Please check this and change accordingly in the text.  Response: again, thank you for spotting this too.  We have adjusted as appropriate.

Reviewer 2 Report

The authors propose a study that has analysed the effects of storm overflows on shellfish waters in England, by processing data from sewerage networks from 4 producers. The topic is interesting, and the study potentially worth of publication, although it is affected by three problems:

1.      There is no proper justification of the study novelty sourcing from a detailed analysis of the state-of-the-art

2.      There is a total lack of a statistical analysis of the data, which could have made the results more robust

3.      In relation to the first point, there are no cross-comparisons with the published literature on the topic in the discussion.

Some minor considerations are reported in the commented MS in attachment.

Author Response

The authors propose a study that has analysed the effects of storm overflows on shellfish waters in England, by processing data from sewerage networks from 4 producers. The topic is interesting, and the study potentially worth of publication, although it is affected by three problems:

  1. There is no proper justification of the study novelty sourcing from a detailed analysis of the state-of-the-art  Response: We have further reviewed the published literature and added some extra justification here but we have struggled to find any peer reviewed publications of direct relevance to our study which assesses spill numbers in shellfisheries.  Happy to adjust further if you have any specific publications in mind?
  2. 2.There is a total lack of a statistical analysis of the data, which could have made the results more robust.  Response: We have now introduced a statistical analysis of the data to assess for differences between water companies and between years.
  3. In relation to the first point, there are no cross-comparisons with the published literature on the topic in the discussion.  Response: As per our response under point 1. we have had difficulty in finding relevant studies for direct comparison but have expanded this section to include discussion of studies of most relevance.  Please advise if you have any other specific studies in mind. 

Some minor considerations are reported in the commented MS in attachment.  Response: Thank you - these have all been addressed as suggested.

Round 2

Reviewer 2 Report

The authors properly addressed all the Reviewers' comments and now the paper is improved.